# Prospective association between depressive symptoms and blood-pressure related outcomes in Kosovo

Katrina A. Obas[1,2], Marek Kwiatkowski[1,2], Ariana Bytyci-Katanolli[1,2], Shukrije Statovci[3], Naim Jerliu[4,5], Qamile Ramadani[6], Nicu Fota[6], Jana Gerold[2,7], Manfred Zahorka[8], Nicole Probst-Hensch[1,2]*

1 Department of Epidemiology and Public Health, Swiss Tropical and Public Health Institute, Allschwill, Switzerland, 2 University of Basel, Basel, Switzerland, 3 University Clinical Center of Kosovo, Prishtina, Kosovo, 4 National Institute of Public Health, Prishtina, Kosovo, 5 University of Prishtina Medical Faculty, Prishtina, Kosovo, 6 Accessible Quality Healthcare Project Implementation Unit, Prishtina, Kosovo, 7 Swiss Centre for International Health, Swiss Tropical and Public Health Institute, Allschwill, Switzerland, 8 OptiMedis AG, Hamburg, Germany

* nicole.probst@swisstph.ch

**Data Availability Statement:** The datasets generated and analyzed during the current study are not publicly available due to non-anonymized

## Abstract

Kosovo has the lowest life expectancy in the Western Balkans, where cardiovascular disease (CVD) accounts for over half of all deaths. Depression also contributes to disability in the country, with a prevalence of moderate to severe symptoms reported as high as 42% in the general population. Although the mechanisms are not yet well understood, evidence suggests that depression is an independent risk factor for CVD. Our study assessed the prospective association between depressive symptoms and blood pressure (BP)-related outcomes among primary healthcare users in Kosovo to understand the role of BP in the relationship between depression and CVD. We included 648 primary healthcare users from the KOSCO study. The presence of depressive symptoms was defined as moderate to very severe depressive symptoms (DASS-21 depressive symptoms score $\geq$14). Multivariable censored regression models assessed prospective associations between baseline depressive symptoms and changes in systolic and diastolic BP while taking hypertension treatment into consideration. Multivariable logistic regression models assessed prospective associations between baseline depressive symptoms and hypertension diagnosis among normotensive patients (n = 226) as well as uncontrolled hypertension in hypertensive patients (n = 422) at follow-up. Depressive symptoms were associated with attenuated diastolic BP ($\beta$ = -2.84, 95%-CI -4.64 to -1.05, p = 0.002) over a year of follow-up in our fully adjusted model, although the association with systolic BP ($\beta$ = -1.98, 95%-CI -5.48 to 1.28, p = 0.23) did not meet statistical significance. We found no statistically significant association of depressive symptoms with hypertension diagnosis among initially normotensive people (OR = 1.68, 95%-CI 0.41 to 6.98, p = 0.48), nor with hypertension control among initially hypertensive people (OR = 0.69, 95%-CI 0.34 to 1.41, p = 0.31). Our findings are not consistent with increased BP as an underlying mechanism between depression and elevated CVD risk and contribute valuable evidence to cardiovascular epidemiology, where the mechanisms between depression, hypertension and CVD are yet to be elucidated.

data in the context of cohort data but are available on reasonable request. The Ethics Committee imposing restrictions is Ethikkommission Nordwest- und Zentralschweiz (Ref. 2018-00994). Data access requests can be sent to: 1. Prof. Dr. Nicole Probst-Hensch Head of Department, Epidemiology and Public Health PI of KOSCO study Swiss Tropical and Public Health Institute nicole.probst@swisstph.ch Direct +41 61 284 83 78 Mobile +41 79 280 34 14 2. Dr. Malin Ziehmer-Wenz Chief Information Security and Data Privacy Officer Swiss Tropical and Public Health Institute malin.ziehmer-wenz@swisstph.ch Direct +41 61 284 87 98.

**Funding:** This work was supported by the Swiss Agency for Development and Cooperation (SDC). The first year of salary for the doctoral studies of KAO (SDC to KAO) and the implementation and running costs of the cohort were funded by SDC (SDC to AQH Project), which is an agency in the federal administration of Switzerland and part of the Federal Department of Foreign Affairs, which are responsible for coordinating Swiss international development projects in Eastern Europe. They are the core funders of the AQH implementation project in which the cohort is embedded. Local SDC representatives were responsible for approving the cohort budget and the study proposal. SDC contributed to the direction of study objectives. The Swiss Tropical and Public Health Institute (Swiss TPH) has internally funded the salary of KAO in the second and third years of doctoral studies (Swiss TPH to KAO). Coauthors associated with Swiss TPH include KAO, JG, ABK, MZ and NPH and contributed to the study in various capacities, specified in the authors' contributions declaration. The Swiss Government Excellence Scholarship for Foreign Scholars and Artists (ESKAS) was awarded to ABK from 2019 to 2022 (Reference number 2019.0234), which funded her doctoral studies salary (ESKAS to ABK).

**Abbreviations:** AQH, Accessible Quality Healthcare Project; BMI, Body Mass Index; CI, Confidence interval; COVID-19, Coronavirus disease; CVD, Cardiovascular disease; DASS-21, 21-item Depression Anxiety Stress Scale; HSCL, Hopkins Symptoms Checklist; IQR, Interquartile range; KOSCO, Kosovo Non-Communicable Disease Cohort; MDD, Major Depressive Disorder; MFMC, Main Family Medicine Centre; mmHg, Millimetres of mercury; OR, Odds ratio; PHC, Primary Healthcare; SD, Standard deviation.

# 1 Introduction

Kosovo has the lowest life expectancy in the Western Balkans [1], and cardiovascular disease is the main contributor to poor health in the country. In 2019, over half (51.1%) of deaths in Kosovo were cardiovascular-related, with women (54.2%) affected more than men (48.5%) [2]. Well-known causes of cardiovascular disease (CVD) include modifiable risk factors such as physical inactivity, smoking, poor diet, alcohol, hypertension and diabetes, as well as genetic predisposition. Baseline findings from the Kosovo non-communicable disease cohort (KOSCO, n = 977) found that poor nutrition (85%), physical inactivity (70%), obesity (53%), and smoking (21%) were common [3].

There is growing evidence that depression is also an independent risk factor for various CVDs. One meta-analysis (n = 80,000, 3–37 years follow-up) found a pooled relative risk of 1.46 (95%-CI 1.37–1.55) [4] for various CVDs, while a more recent meta-analysis (n = 893,850, 2–37 years follow-up) found a pooled relative risk of 1.30 (95%-CI 1.22–1.40) specifically for coronary heart disease and 1.30 (95%-CI, 1.18–1.44) for myocardial infarction [5].

Depression is a leading cause of disability in the world [6]. Understanding the role depression plays in the development and progression of CVD in the Kosovo context is of particular public health relevance given the high prevalence of depression in the country. One 2009 nationally representative study (n = 1161) of persons aged 15 years or older found that 41.7% had moderate to severe depressive symptoms measured by the Hopkins Symptoms Checklist (HSCL) [7]. Another recent study (n = 155) reported 35.6% moderate to severe depressive symptoms during the COVID pandemic in Kosovo [8]. This is well above the mean found in Southeast Asia (16%) [9] but still below the mean found in Africa (45%) [10].

Identifying the underlying mechanisms between depression and CVD is of great value for cardiovascular epidemiology and for integrating mental health and cardiovascular health care, in particular in primary healthcare (PHC). Although the underlying mechanisms are not yet elucidated, the literature has focused on pathways through high blood pressure. Hypertension is a natural target for investigation, given it is the most important risk factor for CVD [11]. Further, when ranked as risk-attributable DALYs, high systolic blood pressure was the leading risk factor globally, accounting for 10.4 million (95%-CI 9.39–11.5) deaths and 218 million (95-%CI 198–237) Disability Adjusted Life-Years [12]. A meta-analysis concluded that depression increases hypertension incidence [13] however the authors cautioned that the limited number of longitudinal studies available may have impacted conclusions. The way hypertension is defined is also an important limitation of the existing literature. Although some studies on the association between depression and hypertension focused on blood pressure measurements alone [14,15], or in combination with hypertension diagnosis or antihypertensive medication use [16–22], several others relied solely on physician-diagnosed hypertension and the use of antihypertension medication to assess the presence of hypertension [23–26]. However, many people remain unaware that they have high blood pressure, especially if they do not experience symptoms and fail to get a diagnosis. Therefore it is important to include blood pressure measurements when defining outcomes related to hypertension.

Looking at changes in blood pressure may improve sensitivity in detecting the effect of depression on increased blood pressure given that some people may fall short of clinical cut-offs of hypertension despite having significant increases in blood pressure over time. The prospective evidence on the effect of depression on the outcome of change in blood pressure is mixed: studies concluded that depression lowered blood pressure [27,28], increased blood pressure [14], or had no effect on blood pressure [16,29]. Further, including change in blood pressure and hypertension diagnosis together in an outcome definition is subject to bias, given that higher primary healthcare utilization unrelated to mental health among people with

depressive symptomology [30] has been observed. Therefore hypertension diagnosis may be a proxy for increased healthcare utilization since healthcare-seeking leads to better detection rather than providing an unbiased assessment of high blood pressure in the population.

In previous work from the authors of this study [31], the prospective association between depression and blood pressure-related outcomes of change in blood pressure and hypertension diagnosis were investigated separately due to the aforementioned limitation. The findings of that study suggest that depressive symptoms among normotensive people both attenuate blood pressure over time yet also increase the likelihood of hypertension diagnosis. The authors, therefore, propose that depression biologically attenuates blood pressure increase over time while increased healthcare utilization among people with depressive symptoms might mediate the association between depression and hypertension diagnosis, however this would need further investigation. This hypothesis however might help explain the conflicting evidence on the association between depression, blood pressure and hypertension, but needs replication.

Our study aims to assess the prospective association between depression and blood pressure related outcomes such as changes in systolic and diastolic blood pressure as well as hypertension diagnosis among normotensive patients and hypertension control among hypertensive patients in Kosovo. The study aims at replicating a previous study conducted in Switzerland [31].

## 2 Methods

### 2.1 Ethics statement

Ethical approvals were obtained from the Ethics Committee Northwest and Central Switzerland (Ref. 2018–00994) and the Kosovo Doctors Chamber (Ref. 11/2019). KOSCO complies with the Declaration of Helsinki. All participants provided informed written consent before participating in any aspect of the KOSCO study.

### 2.2 Study design

We conducted longitudinal analyses using observational data from the Kosovo Non-Communicable Disease Cohort (KOSCO). The KOSCO study is a PHC patient cohort. The main reason for a patient cohort was to enable the evaluation of PHC services given that the KOSCO study is embedded in the Accessible Quality Healthcare Project (AQH), which is devoted to working with local stakeholders to improve the quality of PHC services in Kosovo through health system strengthening strategies.

Data collection occurred approximately every 6 months, starting in 2019 when the cohort was implemented, alternating between in-person interviews and telephone interviews. Due to coronavirus restrictions, the in-person interview planned for follow-up 2 was changed to telephone and the in-person interview was delayed to follow-up 3. At the time of writing, follow-up 4 was completed and follow-up 5 data collection was ongoing. The current study therefore makes use of data from baseline and follow-up 3. The timeline is shown in Table 1. Further details of the KOSCO study are detailed in the study protocol [32].

### 2.3 Setting

The study was conducted in Kosovo, located in the Western Balkans, with 1.8 million inhabitants throughout 38 municipalities over a surface area of nearly 11 000 km$^2$. In Kosovo, the PHC system is divided into three tiers: Each municipality has one Main Family Medicine Center (MFMC), several Family Medicine Centers (FMC) and several Family Medicine Ambulantas (FMA). MFMCs are the largest facilities at the highest level of PHC, which offer more

**Table 1. Timeline of data collection for the Kosovo Non-Communicable Disease Cohort from baseline to follow-up 3.**

| | 2020 | | | | 2020 | | | | 2021 | | | |
|---|---|---|---|---|---|---|---|---|---|---|---|---|
| | Q1 | Q2 | Q3 | Q4 | Q1 | Q2 | Q3 | Q4 | Q1 | Q2 | Q3 | Q4 |
| Baseline | | P | P | P | | | | | | | | |
| Follow-up 1 | | | | T | T | | | | | | | |
| Follow-up 2 | | | | | | T | T | | | | | |
| Follow-up 3 | | | | | | | | P | P | | | |

Q1 –January to March; Q2 –April to June; Q3 –July to September; Q4 –October to December

P–In-person interviews with health assessment; T–Telephone interviews.

services, staff, and medical equipment and therefore have a higher patient flow compared to the second-level FMCs and third-level FMAs. Study sites included MFMCs from the following 12 participating municipalities: Gjakovë, Drenas, Gračanica, Mitrovicë, Junik, Lipjan, Mal-ishevë, Obiliq, Fushe Kosovë, Rahovec, Skenderaj, Vushtrri. MFMCs are PHC facilities, which currently do not offer mental health services. Given that the KOSCO study is embedded within the AQH project, study municipalities were selected based on their partnership with the AQH project.

PHC practices in Kosovo are not yet standardized. One of the AQH interventions for the improvement of PHC services is the implementation of service packages (SPs). An important aspect of the SPs is improving the quality of care by setting standards that should be provided at PHC facilities, based on the WHO PEN Protocols [33] which have been adapted to the Kosovo context by national experts. The SPs ensure a continuum of care with the family physi-cian in a gatekeeper role, where patients who are at risk of developing diabetes or hyperten-sion, or those who have already been diagnosed are referred to a health educator for one-to-one motivational counselling sessions to facilitate behaviour change.

There are only 2.68 psychiatrists and 0.49 psychologists per 100,000 inhabitants [34] com-pared to Switzerland, which has 30 psychiatrists per 100,000 inhabitants [35]. There are eight Community-Based Mental Health Centres in Kosovo, each covering approximately 250,000 inhabitants. The implementation of new community mental health services in Kosovo is still characterized by considerable shortages, including financial and human resources, capacity building, stakeholder involvement and service availability [36]. Additionally, the psychiatry clinic of the University Clinical Centre of Kosovo in Pristina provides the majority of psychiat-ric inpatient capacities of Kosovo (88 beds), and regional psychiatric wards are equipped with 10–25 beds on average, for a total of 166 psychiatric beds excluding Prishtina [34]. This is a psychiatric bed rate (8.3 per 100,000 population) which is roughly 10 times less than in Central European or Scandinavian countries. There are no specialized psychiatric hospitals in Kosovo.

## 2.4 Participants

KOSCO participants were recruited consecutively irrespective of the reason for the PHC visit by trained study nurses as they exited MFMCs. This recruitment method was chosen due to the absence of patient registries which would have enabled randomization. Inclusion criteria for the KOSCO study included: 1) aged 40 years or older, 2) consulted PCH services at the MFMC on the day of recruitment, 3) residence in one of 12 participating municipalities, 4) ability to respond to questions in Albanian or Serbian, 5) no terminal illness such as stage 4 cancer, stage 4 COPD, stage 4 congestive heart failure, and stage 5 chronic kidney disease (CKD) or severe dementia, and 6) living in Kosovo for at least 6 months of the year. Further

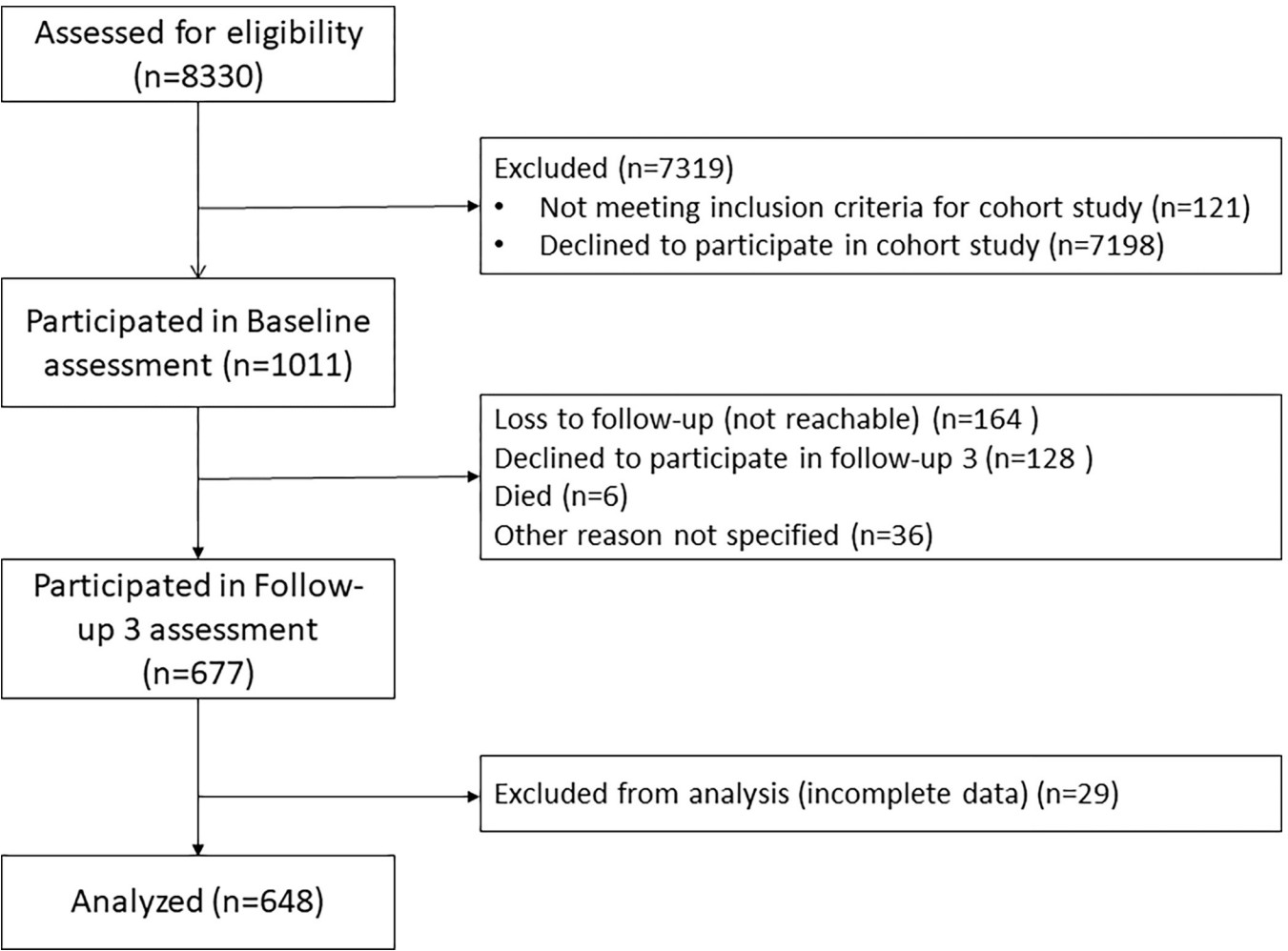

**Fig 1. Flow diagram for inclusion of participants in the current study.**

details of the KOSCO recruitment method and study implementation are described in the study protocol [32].

Participants for the current study met additional criteria: (a) had blood pressure measurement data at baseline (March to November 2019) and follow-up 3 (September 2020 to February 2021), (b) had complete baseline data on depression and confounders, and (c) had complete data on hypertension diagnosis and antihypertensive treatment at baseline and follow-up 3. There were 648 participants from the KOSCO study that met inclusion criteria. **Fig 1** depicts the flow of participants for inclusion in the current study and **Table 2** describes the baseline characteristics of participants included in the current study.

The loss-to-follow-up in the KOSCO study was particularly evident following the outbreak of COVID-19. Retention of over 90% of participants was observed at the first follow-up (October 2019-February 2020) but was reduced to approximately 65% by follow-up 3 (September 2020 to February 2021). The fear of exposure to COVID during study visits, especially before the rolling out of vaccines, was a concern voiced by participants thus affecting the retention of participants and reducing the power of the analyses. However, baseline characteristics were comparable between participants and those who dropped out of the study (refer to **S1 Table**).

**Table 2. Baseline participant characteristics also disaggregated by depressive symptoms status.**

| Sociodemographic factors | All participants (n = 648) | Moderate to very severe depressive symptoms (n = 73 ) | Normal to mild depressive symptoms (n = 575) | |
|---|---|---|---|---|
| | n (%) | n (%) | n (%) | p-value |
| Age, mean (Mean±SD) | 59.4 ±8.9 | 59.4±9.0 | 59.4±8.9 | 0.966[a] |
| Sex, frequency | | | | <0.001[b] |
| Male | 273 (42.1) | 15 (20.6) | 258 (44.9) | |
| Female | 375 (57.9) | 258 (44.9) | 317 (55.1) | |
| Education, frequency | | | | 0.002 [b] |
| Primary school or less | 399 (61.6) | 59 (80.8) | 340 (59.1) | |
| Secondary school | 204 (31.5) | 12 (16.4) | 192 (33.4) | |
| University/College | 45 (6.9) | 2 (2.7) | 43 (7.5) | |
| Work status, frequency | | | | 0.117 [b] |
| Currently working | 119 (18.4) | 6 (8.2) | 113 (19.7) | |
| House person | 303 (46.8) | 37 (50.7) | 266 (46.3) | |
| Retired or disabled | 208 (32.0) | 28 (38.4) | 180 (31.3) | |
| Unemployed | 18 (2.8) | 2 (2.7) | 16 (2.8) | |
| Residence, frequency | | | | 0.025 [b] |
| Rural | 372 (57.4) | 33 (45.2) | 339 (59.0) | |
| Urban | 276 (42.6) | 40 (54.8) | 236 (41.0) | |
| Municipality, frequency | | | | * |
| Drenas | 63 (9.7) | 17 (23.3) | 46 (8.0) | |
| Fushe Kosova | 64 (9.9) | 12 (16.4) | 52 (9.0) | |
| Gjakova | 51 (7.9) | 5 (6.9) | 46 (8.0) | |
| Gračanica | 36 (5.6) | 3 (4.1) | 33 (5.7) | |
| Junik | 11 (1.7) | 0 (0.0) | 11 (1.9) | |
| Lipjan | 106 (16.4) | 4 (5.5) | 102 (17.7) | |
| Malisheva | 51 (7.9) | 0 (0.0) | 51 (8.9) | |
| Mitrovica | 65 (10.0) | 13 (17.8) | 52 (9.0) | |
| Obiliq | 37 (5.7) | 2 (2.7) | 35 (6.1) | |
| Rahovec | 53 (8.2) | 1 (1.4) | 52 (9.0) | |
| Skenderaj | 69 (10.7) | 12 (16.4) | 57 (9.9) | |
| Vushtrri | 42 (6.5) | 4 (5.5) | 38 (6.6) | |
| Ethnicity, frequency | | | | <0.001 [b] |
| Albanian | 589 (90.9) | 63 (86.3) | 526 (91.5) | |
| Serbian | 34 (5.2) | 1 (1.4) | 33 (5.7) | |
| Roma, Ashkali, Egyptian, Other | 25 (3.9) | 9 (12.3) | 16 (2.8) | |
| Main Family Medicine Center visits in the last 6 months, median (IQR) | 3 (2-6) | 6 (3-10) | 3 (2-6) | 0.007 [a] |
| Smoking, frequency | | | | 0.022 [b] |
| Never or ex-smoker | 521 (80.4) | 66 (90.4) | 455 (79.1) | |
| Current smoker | 127 (19.6) | 7 (9.6) | 120 (20.9) | |
| Physical activity, frequency | | | | 0.639 [b] |
| Sufficiently active | 211 (32.6) | 22 (30.1) | 189 (32.9) | |
| Insufficiently active | 437 (67.4) | 51 (69.9) | 386 (67.1) | |
| Alcohol, frequency | | | | 0.039 [b] |
| No alcohol in past 30 days | 616 (95.1) | 73 (100.0) | 543 (94.4) | |
| Consumed alcohol in past 30 days | 32 (4.9) | 0 (0.0) | 32 (5.6) | |

(*Continued*)

**Table 2.** (Continued)

| Sociodemographic factors | All participants (n = 648) | Moderate to very severe depressive symptoms (n = 73 ) | Normal to mild depressive symptoms (n = 575) | |
|---|---|---|---|---|
| | n (%) | n (%) | n (%) | p-value |
| Nutrition, frequency | | | | 0.039 [b] |
| Adequate nutrition | 98 (15.1) | 17 (23.3) | 81 (14.1) | |
| Poor nutrition | 550 (84.9) | 56 (76.7) | 494 (85.9) | |
| Sleep, frequency | | | | <0.001 [c] |
| Very good | 175 (27.0) | 7 (9.6) | 168 (29.2) | |
| Fairly good | 236 (36.4) | 19 (26.0) | 217 (37.7) | |
| Fairly bad | 173 (26.7) | 25 (34.3) | 148 (25.7) | |
| Very bad | 64 (9.9) | 22 (30.1) | 42 (7.3) | |
| Obesity, frequency | | | | 0.040 [b] |
| BMI <30 | 286 (44.1) | 24 (32.9) | 262 (45.6) | |
| BMI ≥ 30 | 362 (55.9) | 49 (67.1) | 313 (54.4) | |
| Systolic blood pressure (mmHg), mean (Mean±SD) | 135.7±17.9 | 137.4±20.1 | 135.4±17.7 | 0.383 [a] |
| Change in systolic blood pressure, mean (Mean±SD) | 2.2±14.0 | 0.5±17.3 | 2.5±13.5 | 0.270 [a] |
| Diastolic blood pressure (mmHg), mean (Mean±SD) | 86.4±9.9 | 88.8±10.8 | 86.1±9.8 | 0.029 [a] |
| Change in diastolic blood pressure, mean (Mean±SD) | 0.4±7.8 | -2.6±9.0 | 0.7±7.6 | <0.001 [a] |
| Hypertension, frequency | | | | 0.048 [b] |
| Never diagnosed | 246 (38.0) | 20 (27.4) | 226 (39.3) | |
| Diagnosed | 402 (62.0) | 53 (72.6) | 349 (60.7) | |
| Antihypertensive treatment, frequency | | | | 0.107 [b] |
| Not taking | 359 (55.4) | 34 (46.6) | 325 (56.5) | |
| Taking | 289 (44.6) | 39 (53.4) | 250 (43.5) | |
| Depressive symptoms at baseline, frequency | | | | N/A |
| Normal-mild (DASS <14) | 575 (88.7) | 0 (0.0) | 575 (100.0) | |
| Moderate to very severe (DASS≥14) | 73 (11.3) | 73 (100.0) | 0 (0.0) | |

mmHg: millimetres of mercury, DASS: Depression Anxiety Stress Scale, BMI: body mass index, SD: standard deviation, IQR: interquartile range. Normal to mild depressive symptoms if depression subscale of 21-item Depression Anxiety Stress Scale score was <14. Moderate to very severe depressive symptoms if depression subscale of 21-item Depression Anxiety Stress Scale score was ≥14. a t-test; b Chi-square test; cKruskall-Wallis test; *Fisher's exact test not possible due to high number of categories

## 2.5 Variables

**2.5.1 Change in systolic and diastolic blood pressure.** Because clinically significant increases in blood pressure may fall short of specified cut-off criteria for hypertension, we calculated changes in blood pressure and used the change scores as a continuous outcome variable for systolic and diastolic blood pressure.

Systolic and diastolic blood pressure (in mmHg) was measured three times, at least three minutes apart, after sitting quietly for about 10 minutes, using an M3 model Omron blood pressure monitor (Omron Healthcare, Switzerland). The research nurses placed the blood pressure cuff two centimetres above the elbow on the bare left upper arm (in the case of arteriovenous fistula, radiotherapy or removal of lymph nodes in the armpit of the left arm, the right arm was used) of the seated participant and elevated the arm on the table to the level of the fourth intercostal space. Participants provided additional written consent that they would like to be informed by the study nurse in case high blood pressure was detected throughout the study.

Changes in systolic and diastolic blood pressure were calculated by subtracting blood pressure at baseline from blood pressure at follow-up 3. Because follow-up time varied from 0.9 to 1.9 years between participants, we standardized the change in blood pressure as a unit of change over 1 year by dividing it by the follow-up time in years. A positive censored regression coefficient for moderate to very severe depressive symptoms indicates a larger increase in blood pressure over time compared to the reference group, while a negative coefficient represents a smaller increase in blood pressure over time compared to the reference group. A negative coefficient should not be interpreted as a decrease in blood pressure over time.

**2.5.2 Incident hypertension diagnosis.** Incident hypertension diagnosis was considered as newly self-reported physician-diagnosed hypertension or hypertension treatment at follow-up 3, which were determined from interview questions about medication and disease diagnoses. Incident hypertension diagnosis occurred when a participant had no self-reported diagnosis nor treatment for hypertension at baseline (regardless of the blood pressure measurement), and hypertension diagnosis or treatment at follow-up 3.

**2.5.3 Uncontrolled hypertension.** Keeping blood pressure within normal limits after the diagnosis of hypertension through lifestyle changes and medication is important to avoid disease progression. Uncontrolled hypertension was considered as systolic blood pressure ≥140 mm Hg or diastolic blood pressure ≥90 mm Hg at follow-up 3 among people previously diagnosed with hypertension at baseline.

**2.5.4 Depressive symptoms.** The 21-item Depression Anxiety Stress Scale (DASS-21) was designed to measure current levels of depressive symptomology and should be used as a screening tool rather than a diagnostic tool for Major Depressive Disorder (MDD). Depressive symptoms were measured by interview using the DASS-21 [37,38] questionnaire which includes a subscale for depressive symptoms containing seven items scored on a 4-point Likert scale ranging from 0 (did not apply to me at all) to 3 (applied to me very much), and multiplied by 2. The presence of depressive symptoms was defined as moderate to very severe depressive symptoms (DASS-21 depressive symptoms score of ≥14). The severity of depressive symptoms is categorized as follows: DASS-21 score 0–9 (normal), 10–13 (mild), 14–20 (moderate), 21–27 (severe), and over 28 (very severe).

**2.5.5 Statistical methods.** *2.5.5.1 Depression and change in blood pressure.* Many study participants who were normotensive at baseline developed hypertension and were prescribed antihypertensive medication before their follow-up health assessments. Other participants who took antihypertensive medication at baseline stopped treatment by follow-up 3 for various reasons. Accounting for the effect of this treatment on the measured blood pressure is of vital importance in statistical analyses of blood pressure. Several analytical strategies have been evaluated in a simulation study [39], which recommended the use of censored normal regression. In this approach, the blood pressure change for all participants newly diagnosed or treated for hypertension at follow-up 3 was right-censored at the measured value. This is equivalent to assuming that had these participants not received the diagnosis and/or treatment, their blood pressure at the follow-up assessment would have been equal to or greater than the measured value. The blood pressure change for all participants who were taking antihypertensive treatment at baseline but stopped treatment at follow-up 3 was left-censored at the measured value. The coefficients fitted by censored regression have the same familiar interpretation as those from ordinary linear regression.

We fitted separate mixed censored regression models for change in systolic and diastolic blood pressure. We termed models that were adjusted only for the confounders "minimally adjusted", and models further controlled for the suspected mediators "fully adjusted". Effect modification by sex was assessed by introducing the appropriate interaction term. We adjusted our models for baseline blood pressure, because of evidence of severe regression to the mean.

As this adjustment may introduce bias due to the potential fluctuations of measurements [40], we report also effect estimates from otherwise identical models without baseline adjustment in the **S2 Table.**

We considered the following confounders based on prior knowledge in all models: age (in years), sex (male, female), highest level of education completed (primary school or less, secondary school, university/college or more), work (working, home-person, retired/disabled, unemployed), urban-rural classification (rural, urban), and municipality as a random effect (Gjakovë, Drenas, Graçanicë, Mitrovicë, Junik, Lipjan, Malishevë, Obiliq, Fushe Kosovë, Rahovec, Skenderaj, Vushtrri), and ethnicity (Albanian, Serbian, Roma/Ashkali/Egyptian/Other).

We considered the following factors as potential mediators of the association between depression and change in blood pressure based on previous evidence [41–44]: *smoking status* (current smoker), *physical inactivity* (<150 min of moderate-intensity physical activity per week, or <75 min of vigorous-intensity physical activity per week, or less than an equivalent combination of moderate-intensity and vigorous-intensity activity); *poor nutrition* (<5 fruits and/or vegetables per day), *alcohol consumption* (any alcohol in the last 30 days), *obesity* (BMI≥30), *heart rate* (beats per minute, and the *number of visits to an MFMC in the last 6 months)*. Each of these mediators was also assessed individually for mediating effect by introducing each variable one at a time in the minimally adjusted model. These results are provided in **S3 Table**.

*2.5.5.2 Depression and incident hypertension diagnosis as well as uncontrolled hypertension.* The prospective associations between baseline depression and the outcomes of incident hypertension diagnosis and uncontrolled hypertension were assessed with mixed multivariable logistic regression models. We included the same set of covariates as the fully adjusted models for change in blood pressure and additional adjustment for time of follow-up. Effect modification by sex was assessed by introducing the appropriate interaction term (see **S4 Table**).

Analyses were performed with Stata statistical software, release 16.

## 3 Results

### 3.1 Baseline depression and change in systolic and diastolic blood pressure

**Table 3** shows the findings of the minimally and fully adjusted models on the prospective association between depressive symptoms (binary, severity category) and change in systolic and diastolic blood pressure over one year of follow-up. Depressive symptoms as a binary predictor were associated with attenuated diastolic blood pressure increase (β = -2.84, 95%-CI -4.64 to -1.05, p = 0.002) over a year of follow-up in our fully adjusted models, while the attenuation of systolic blood pressure increase (β = -1.98, 95%-CI -5.25 to 1.28, p = 0.233) was not statistically significant. When considering the severity of depressive symptoms as categories defined by Lovibond and Lovibond [37], we found that only moderate levels of depressive symptoms were predictive of attenuation of diastolic blood pressure increase in the minimally and fully adjusted models. No other findings were statistically significant. No effect modification by sex was observed (see **S4 Table**) and no evidence of mediation by lifestyle risk factors was found.

### 3.2 Baseline depression and incidence of hypertension diagnosis as well as uncontrolled hypertension

**Table 4** shows the findings of the minimally and fully adjusted models of the prospective association between depressive symptoms and incident hypertension diagnosis among normotensive patients as well as uncontrolled hypertension among hypertensive patients. Although no

**Table 3. Prospective association between depression and change in systolic and diastolic blood pressure per year.**

| | Change in systolic blood pressure | | | | | | Change in diastolic blood pressure | | | | | |
| --- | --- | --- | --- | --- | --- | --- | --- | --- | --- | --- | --- | --- |
| | Minimally [a] adjusted | | | Fully [b] adjusted | | | Minimally [a] adjusted | | | Fully [b] adjusted | | |
| | Coef | 95%-CI | p-value | Coef | 95%-CI | p-value | Coef | 95%-CI | p-value | Coef | 95%-CI | p-value |
| **Depression** | | | | | | | | | | | | |
| Normal to mild depressive symptoms (DASS<14) | (Ref) | | | (Ref) | | | (Ref) | | | (Ref) | | |
| Moderate to very severe depressive symptoms (DASS ≥14) | -2.31 | (-5.48 to 0.87) | 0.155 | -1.98 | (-5.25 to 1.28) | 0.233 | -2.93 | (-4.68 to -1.18) | 0.001 | -2.84 | (-4.64 to -1.05) | 0.002 |
| **Depression severity (categorical)** | | | | | | | | | | | | |
| Normal (DASS-21 score 0-9) | (Ref) | | | (Ref) | | | (Ref) | | | (Ref) | | |
| Mild (DASS-21 score 10-13) | 1.08 | (-2.65 to 4.80) | 0.571 | 1.64 | (-2.11 to 5.40) | 0.391 | 1.41 | (-0.64 to 3.46) | 0.178 | 1.86 | (-0.19 to 3.92) | 0.075 |
| Moderate (DASS-21 score 14-20) | -3.41 | (-7.16 to 0.35) | 0.075 | -3.14 | (-6.96 to 0.67) | 0.107 | -3.43 | (-5.50 to -1.35) | 0.001 | -3.41 | (-5.51 to -1.32) | 0.001 |
| Severe (DASS-21 score 21-27) | -1.15 | (-8.47 to 6.18) | 0.759 | -0.23 | (-7.56 to 7.10) | 0.951 | -2.67 | (-6.72 to 1.38) | 0.196 | -2.29 | (-6.32 to 1.73) | 0.265 |
| Very severe (DASS-21 score ≥ 28) | 2.24 | (-5.25 to 9.73) | 0.557 | 2.91 | (-4.66 to 10.49) | 0.451 | 0.20 | (-3.92 to 4.31) | 0.925 | 0.92 | (-3.21 to 5.05) | 0.663 |

Results from multivariable censored regression models. [a] minimally adjusted: age (in years), sex (male, female), highest level of education completed (primary school or less, secondary school, university/college or more), work (working, home-person, retired/disabled, unemployed), urban-rural classification (rural, urban), ethnicity (Albanian, Serbian, Roma/Ashkali/Egyptian/Other), baseline systolic blood pressure, baseline diastolic blood pressure. [b] fully adjusted: minimally adjusted covariates and additionally, smoking status (current smoker), physical inactivity (<150 min of moderate-intensity physical activity per week, or <75 min of vigorous-intensity physical activity per week, or less than an equivalent combination of moderate-intensity and vigorous-intensity activity; poor nutrition (<5 fruits and/or vegetables per day), alcohol consumption (any alcohol in the last 30 days), obesity (BMI≥30), heart rate (beats per minutes), number of main family medicine center visits in the last 6 months. DASS-21: 21-item Depression Anxiety Stress Scale; Ref: Reference group; mmHg: millimetres of mercury

finding met statistical significance, depressive symptoms as a binary predictor were associated with a suggestive increase in odds of hypertension diagnosis (OR = 1.68, 95%-CI 0.41, 6.98, p = 0.475) and decrease in odds of uncontrolled hypertension (OR = 0.69, 95%-CI 0.34, 1.40, p = 0.302) in the fully adjusted models.

# 4 Discussion

## 4.1 Main findings

This is the first study to the best of our knowledge to investigate the effect of mental health on blood pressure-related outcomes in Kosovo. We purposefully separately assessed the outcomes of blood pressure change, hypertension diagnosis, and uncontrolled hypertension, in contrast to studies which used a mixture of diagnosis, medication or blood pressure measurement cut-off in the outcome definition. We wanted to disentangle outcomes related to biological processes (blood pressure change) versus behavioural factors (hypertension diagnosis and control), which may help elucidate the mixed evidence on the association between depression and blood pressure.

On the one hand, we found suggestive evidence that baseline moderate to very severe depressive symptoms were associated with an attenuation of systolic and diastolic blood pressure increase over a year. Blood pressure tends to increase with age. On the other hand, we also found suggestive evidence that baseline moderate to very severe depressive symptoms were associated with an increase in odds of hypertension diagnosis among initially normotensive people and a decrease in odds of uncontrolled hypertension among initially hypertensive

**Table 4. Prospective association between depression and incident hypertension diagnosis as well as uncontrolled hypertension.**

| | Incident hypertension diagnosis (n = 226) | | | | | | Uncontrolled hypertension (n = 422) | | | | | |
|---|---|---|---|---|---|---|---|---|---|---|---|---|
| | Minimally [a] adjusted | | | Fully [b] adjusted | | | Minimally [a] adjusted | | | Fully [b] adjusted | | |
| | OR | 95%-CI | p-value | OR | 95%-CI | p-value | OR | 95%-CI | p-value | OR | 95%-CI | p-value |
| **Depression (binary)** | | | | | | | | | | | | |
| Normal to mild depressive symptoms (DASS<14) | (Ref) | | | (Ref) | | | (Ref) | | | (Ref) | | |
| Moderate to very severe depressive symptoms (DASS ≥14) | 1.22 | (0.37 to 4.01) | 0.745 | 1.68 | (0.41 to 6.98) | 0.475 | 0.66 | (0.34 to 1.28) | 0.219 | 0.69 | (0.34 to 1.41) | 0.309 |
| **Depression severity (categorical)** | | | | | | | | | | | | |
| Normal (DASS-21 score 0–9) | (Ref) | | | (Ref) | | | (Ref) | | | (Ref) | | |
| Mild (DASS-21 score 10–13) | 5.17 | (1.10 to 24.28) | 0.037 | 6.87 | (1.32 to 35.68) | 0.022 | 0.74 | (0.35 to 1.59) | 0.439 | 0.97 | (0.44 to 2.13) | 0.940 |
| Moderate (DASS-21 score 14–20) | 1.79 | (0.49 to 6.52) | 0.375 | 2.50 | (0.55 to 11.49) | 0.238 | 0.54 | (0.24 to 1.21) | 0.138 | 0.55 | (0.24 to 1.28) | 0.168 |
| Severe (DASS-21 score 21–27) | 1 | - | - | 1 | - | - | 0.66 | (0.15 to 2.90) | 0.579 | 0.83 | (0.17 to 4.09) | 0.836 |
| Very severe (DASS-21 score ≥ 28) | 0.81 | (0.03 to 21.72) | 0.900 | 1.68 | (0.05 to 53.36) | 0.768 | 1.14 | (0.23 to 5.81) | 0.871 | 1.47 | (0.27 to 8.09) | 0.658 |

Results from multivariable logistic regression models. [a] minimally adjusted: age (in years), sex (male, female), highest level of education completed (primary school or less, secondary school, university/college or more), work (working, home-person, retired/disabled, unemployed), urban-rural classification (rural, urban), ethnicity (Albanian, Serbian, Roma/Ashkali/Egyptian/Other), baseline systolic blood pressure, baseline diastolic blood pressure. [b] fully adjusted: minimally adjusted covariates and additionally, smoking status (current smoker), physical inactivity (<150 min of moderate-intensity physical activity per week, or <75 min of vigorous-intensity physical activity per week, or less than an equivalent combination of moderate-intensity and vigorous-intensity activity; poor nutrition (<5 fruits and/or vegetables per day), alcohol consumption (any alcohol in the last 30 days), obesity (BMI≥30), heart rate (beats per minutes), baseline number of main family medicine center visits in the last 6 months, follow-up time (years). DASS-21: 21-item Depression Anxiety Stress Scale, Ref: reference group; mmHg: millimetres of mercury

people. Only the findings with attenuated diastolic blood pressure met statistical significance. Overall, the direction of the results from this replication study agrees with those from a Swiss cohort with longer-term follow-up [31]. Although the findings taken together appear contradicting, they expose two probable differing effects of depression: depression attenuates blood pressure, which is in line with the monoamine theory of depression, while depression also likely improves the diagnosis and treatment of hypertension related to increased healthcare utilization among depressed people.

## 4.2 Depression and change in systolic and diastolic blood pressure

The finding that depression is longitudinally associated with attenuated blood pressure increase is supported by one study that also included both normotensive and hypertensive people over an 11-year follow-up [27] and another study with only normotensive people [31]. The attenuation of the increase in blood pressure in people with depressive symptoms could be due to several factors including biological, social, and healthcare factors. First, our findings are consistent with the monoamine theory of depression [45], which suggests that the pathogenesis of depression is a depletion of serotonin, norepinephrine and dopamine, which are important for raising blood pressure. This means that in the instance of depletion of monoamines, both depressive symptoms and lower blood pressure can be observed. However, there exist many other theories of the pathogenesis of depression which would argue more for an effect of an increase in blood pressure. One main theory includes autonomic nervous system dysfunction, resulting in decreased heart rate variability and increased heart rate [46] which in turn lead to an increase in blood pressure. There is a risk that we controlled for this effect by

including heart rate as a mediator, however there were no notable differences in effect when it was accounted for individually as seen in **S3 Table**. Several other theories are discussed elsewhere [41,47–49]. It should also be noted that neurobiological evidence on its effect on blood pressure is mostly theoretical and has not sufficiently been observed in human studies. Secondly, given the limited research in the Kosovo context, we may have inadvertently omitted important confounding sociodemographic factors for this population that may bias the association between depressive symptoms and blood pressure change. Finally, given that care standards are still being developed in the country, it is highly probable that patients are being treated inconsistently. The use of censored regression to account for treatment effects may not have sufficiently accounted for this. The AQH project is supporting the implementation of the World Health Organization WHO package of essential noncommunicable (PEN) disease interventions for primary health care, however the adoption of the PEN protocols is only in its early phases. The evaluation of the effect modification by different care standards was thus beyond the scope of this study.

A further investigation into the severity response of depression on blood pressure change is warranted, given our observation that moderate depressive symptoms saw the largest effect of diastolic blood pressure attenuation with a decreasing attenuation in severe depressive symptoms and even a positive change in blood pressure in very severe depressive symptoms, although only the level of moderate depressive symptoms on diastolic blood pressure was statistically significant.

We accounted for lifestyle risk factors such as smoking, alcohol consumption, poor diet, physical inactivity and obesity in our analyses because they are associated with depression [41,42] and are risk factors for hypertension and cardiovascular disease. There were only minimal discrepancies between our minimally and fully adjusted models, suggesting that they neither confounded nor mediated the observed associations in major ways (see **S2 Table**). This is only true under the assumption that these variables were captured adequately and with little reporting bias. We previously observed gender differences between lifestyle factors of this population [3], but the sample size limitations did not allow for taking these interrelations between covariates into consideration. Our finding of little change in effect size from non-adjusted and adjusted lifestyle models is consistent with a review [41], suggesting that the increased risk for CVD mortality is not simply due to differences in lifestyle risk factors.

## 4.3 Depression and incident of hypertension diagnosis

We found suggestive evidence that initially normotensive people with depressive symptoms were more likely to be diagnosed with hypertension by a general practitioner in comparison to non-depressed, although our findings did not meet statistical significance. While the direction of these estimates is entirely consistent with the findings of other studies with a similar definition of hypertension diagnosis as in our study [23–26,31], they are also consistent with the null hypothesis of no association. We ascribe this large uncertainty to the small sample sizes and the short follow-up in this current study. As previously mentioned, the outcome of hypertension diagnosis without the consideration of blood pressure measurement (and therefore underdiagnosis) has implications for the interpretations of the findings. It should be interpreted to be a proxy for the act of receiving a diagnosis of hypertension, which is influenced by several factors related to behaviours. One explanation for our findings is related to healthcare-seeking behaviour. A Swiss study [31] hypothesized that the increased risk of hypertension diagnosed among people with depression was potentially related to their higher healthcare-seeking behaviour, but the study could not verify this factor. Higher health service utilization among depressed people was observed in other studies [30,44] as well as in our sample (see

Table 2). As healthcare utilization was included among covariates of our fully-adjusted model, we may have accounted for its mediating effect and this may contribute to non-statistically significant findings. Another explanation for our findings may be related to a higher prevalence of disease comorbidity [50] or lifestyle risk factors related to other chronic diseases [41,42] among people living with depression, leading to increased hypertension screening. Although lifestyle risk factors were accounted for in our study, comorbidity was not. Nevertheless, both minimally and fully adjusted models did not meet statistical significance.

## 4.4 Depression and uncontrolled hypertension

We also found suggestive evidence that among initially hypertensive subjects, depressive symptoms were associated with a decrease in odds for uncontrolled hypertension, although again these findings did not meet statistical significance. In other words, people with depressive symptoms might be more likely to have their blood pressure under control. This is in contrast to several other studies [51–56]. However, a recent cross-sectional study in the Netherlands (HELIUS study, n = 21 363) found that hypertension control among depressed people differed between ethnic groups [57]. This supports the importance to understand the association in the Kosovo context, especially given that we observed a much higher prevalence of depression among the Roma, Ashkali, and Egyptian ethnic minority group and poorer hypertension control was observed in ethnic minority groups as seen in the United States [58,59] who might have lower access to healthcare. Lower odds of uncontrolled hypertension among the depressed is also in keeping with the hypothesis of higher health-seeking behaviour among people with depression, i.e. as depressed people use more healthcare, they are exposed to treatment that can help reduce blood pressure. However, we accounted for it in our fully adjusted models, possibly explaining findings of no statistically significant findings. Nevertheless, we consider the small sample size and short follow-up time to play an important role (n = 422).

## 4.5 Implications for clinicians and policymakers

Clinicians should interpret our findings carefully. At first glance, one might be inclined to see our findings as promoting depression as a protective factor against hypertension. However, blood pressure increased over time in both those with and without depressive symptomology. Therefore people with depression can still develop hypertension, especially if their baseline blood pressure is borderline hypertensive.

Currently there is no practice in Kosovo PHC to screen for hypertension among depressed people or vice versa. Given the strong stigma of mental illness still present in the country [60] as in many other countries, caution should be taken in regards to widespread screening of depression because it may deter patients from seeking PHC services. One must also consider the ethical implications of screening for depression. A diagnosis should be followed by adequate treatment. Mental health services, especially at the community level in Kosovo are still in development and need to be strengthened [34,36], which require a considerable amount of human and financial resources. A more pressing recommendation would be for policymakers to invest in contextually-acceptable already-validated inexpensive interventions promoting mental health.

## 4.6 Strengths and limitations

This is the first study to the best of our knowledge to assess the effect of depression on blood pressure-related outcomes in Kosovo, a context where both depression and CVD are highly prevalent. It contributes valuable findings on chronic disease epidemiology.

The current study is one of the first publications on KOSCO findings, which was implemented in 2019. Therefore the mean follow-up time for the analyses conducted for this study was 1.3 years (min 0.9 years–max 1.9 years), and may not be sufficient follow-up time to assess change in blood pressure at the population level. Follow-up of the KOSCO participants is anticipated for 5 years which would allow greater follow-up time to observe changes in blood pressure.

As our study is a PHC patient-based cohort, the study is limited in its generalizability to the general population. Further, selection bias must be considered when interpreting our findings, given that most patients approached to be included in the study declined (seen in Fig 1).

The relatively small sample size is a limitation of our study. The loss to follow-up is summarized in the methods indicating retention of 67% from baseline to follow-up 3. Nevertheless, an assessment of baseline characteristics between those lost to follow-up and those included in the study indicates no meaningful differences.

Loss to follow-up was larger after the onset of the COVID-19 pandemic which is well known to impact depression symptoms [61]. Differential loss-to-follow-up of hypertensive participants with elevated symptoms of depression may have introduced bias. But no difference in follow-up participation according to either hypertension-related factors or symptoms of depression was observed.

### 4.7 Conclusion

In Kosovo, depression is associated with an attenuation of diastolic blood pressure increase and we have suggestive evidence of better detection and control of hypertension. Given that the mechanisms between depression, hypertension and CVD are yet to be elucidated, this study contributes valuable evidence towards cardiovascular epidemiology. Larger studies on the prospective association between depression and blood pressure-related outcomes are warranted that attempt to address the full complexity of potentially interacting pathways.

## Supporting information

**S1 Table. Baseline characteristics disaggregated by participant and non-participant status.** mmHg: Millimetres of mercury, DASS: Depression Anxiety Stress Scale, BMI: Body mass index, SD: Standard deviation, IQR: Interquartile range. Normal to mild depressive symptoms if depression subscale of 21-item Depression Anxiety Stress Scale score was <14. Moderate to very severe depressive symptoms if depression subscale of 21-item Depression Anxiety Stress Scale score was ≥14.
(DOCX)

**S2 Table. Prospective association between depression and change in systolic and diastolic blood pressure per year, without adjustment for baseline systolic and diastolic blood pressure.** Results from multivariable censored regression models. [a] Minimally adjusted: Age (in years), sex (male, female), highest level of education completed (primary school or less, secondary school, university/college or more), work (working, home-person, retired/disabled, unemployed), urban-rural classification (rural, urban), ethnicity (Albanian, Serbian, Roma/Ashkali/Egyptian/Other). [b] Fully adjusted: Minimally adjusted covariates and additionally, smoking status (current smoker), physical inactivity (<150 min of moderate-intensity physical activity per week, or <75 min of vigorous-intensity physical activity per week, or less than an equivalent combination of moderate-intensity. DASS-21: 21-item Depression Anxiety Stress Scale, Ref: Reference group.
(DOCX)

**S3 Table. Prospective association between depression and change in systolic and diastolic blood pressure, minimally adjusted model and introduction of potential mediators individually.**
(DOCX)

**S4 Table. P-values of the interaction term between depression and sex in main fully adjusted models.**
(DOCX)

## Acknowledgments

The implementation of the cohort study protocol and prioritization of study objectives would not be possible without the support of the Project Implementation Unit of the AQH project. Staff from 12 participating Main Family Medicine Centers (MFMCs) supported the study along with MFMC directors in the following municipalities: Fushë Kosovë, Gjakovë, Drenas, Gracanica, Junik, Lipjan, Malishevë, Mitrovicë, Obiliq, Rahovec, Skenderaj, and Vushtrri. The KOSCO research nurses were instrumental in the success of recruitment and data collection: Tevide Bllaca, Arizona Igrishta, Selvete Zyberaj, Shqipe Agushi, Alma Stojanovic, Fatime Zeneli, Artina Igrishta, Arizona Krasniqi and Nikola Sekulic.

## Author Contributions

**Conceptualization:** Marek Kwiatkowski, Shukrije Statovci, Naim Jerliu, Qamile Ramadani, Nicu Fota, Jana Gerold, Manfred Zahorka, Nicole Probst-Hensch.

**Data curation:** Katrina A. Obas, Ariana Bytyci-Katanolli.

**Formal analysis:** Katrina A. Obas, Marek Kwiatkowski.

**Funding acquisition:** Manfred Zahorka, Nicole Probst-Hensch.

**Investigation:** Katrina A. Obas, Nicole Probst-Hensch.

**Methodology:** Katrina A. Obas, Marek Kwiatkowski, Nicole Probst-Hensch.

**Project administration:** Katrina A. Obas.

**Resources:** Nicole Probst-Hensch.

**Supervision:** Marek Kwiatkowski, Nicole Probst-Hensch.

**Validation:** Nicole Probst-Hensch.

**Writing – original draft:** Katrina A. Obas.

**Writing – review & editing:** Katrina A. Obas, Marek Kwiatkowski, Ariana Bytyci-Katanolli, Shukrije Statovci, Naim Jerliu, Qamile Ramadani, Nicu Fota, Jana Gerold, Manfred Zahorka, Nicole Probst-Hensch.

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
