## [Editor Report · Decision Letter 0]

27 Jun 2022

PGPH-D-22-00718

Prospective association between depressive symptoms and blood-pressure related outcomes in Kosovo

Dear Dr. Probst-Hensch,

Thank you for submitting your manuscript to PLOS Global Public Health. After careful consideration, we feel that it has merit but does not fully meet PLOS Global Public Health’s publication criteria as it currently stands. Therefore, we invite you to submit a revised version of the manuscript that addresses the points raised during the review process.

Please submit your revised manuscript by . If you will need more time than this to complete your revisions, please reply to this message or contact the journal office at globalpubhealth@plos.org. Please include the following items when submitting your revised manuscript:

We look forward to receiving your revised manuscript.

Kind regards,

Anil Gumber, Ph.D.

Academic Editor

Journal Requirements:

b. State what role the funders took in the study. If the funders had no role in your study, please state: “The funders had no role in study design, data collection and analysis, decision to publish, or preparation of the manuscript.

2. Please update the Funding Information section in the system and ensure that it matches with Financial Disclosure Statement.

3. In the online submission form, you indicated that "The datasets generated and analyzed during the current study are not publicly available due to non-anonymized data in the context of cohort data but are available from the corresponding author on reasonable request.". All PLOS journals now require all data underlying the findings described in their manuscript to be freely available to other researchers, either 1. In a public repository, 2. Within the manuscript itself, or 3. Uploaded as supplementary information.

4. Please provide separate figure files in .tif or .eps format and removed from the manuscript file.

5. We have noticed that you have uploaded Supporting Information files, but you have not included a list of legends. Please add a full list of legends for your Supporting Information files after the references list. 

Additional Editor Comments (if provided):

The paper is well written. Abstract need revision because non-significant relationships can't be emphasized. Tables need statistical significance level to understand 95% CI is in the significant range or not. Statistical significance by Sociodemographic factors between two groups are required (Table 1). Adjusted models needs to be run separately for covariates/contextual factors and confounders/mediating factors to see some independent effect on the model and might show some significance levels. It is not clear to the reader what was done during the follow-up period (any intervention). Also it is not clear what happened to DASS scale scores over-time. Whether there is any cross-over between mild/moderate vs severe Depression groups over time.
---

## [Decision Letter · Decision Letter 1]

6 Nov 2022

PGPH-D-22-00718R1

Prospective association between depressive symptoms and blood-pressure related outcomes in Kosovo

Dear Dr. Probst-Hensch,

Thank you for submitting your manuscript to PLOS Global Public Health. After careful consideration, we feel that it has merit but does not fully meet PLOS Global Public Health’s publication criteria as it currently stands. Therefore, we invite you to submit a revised version of the manuscript that addresses the points raised during the review process.

The manuscript has been evaluated by three reviewers, and the comments from reviewers 1 and 2 are available below, and in the attached documents. Reviewer 3 notes that the revised manuscript adequately meets all of the recommendations from the original review. 

The reviewers have raised a number of concerns that need attention. 

Could you please revise the manuscript to carefully address the concerns raised?

We look forward to receiving your revised manuscript.

Kind regards,

Steve Zimmerman, PhD

PLOS Staff Editor

Journal Requirements:

2. In the online submission form, you indicated that "The datasets generated and analyzed during the current study are not publicly available due to non-anonymized data in the context of cohort data but are available on reasonable request". All PLOS journals now require all data underlying the findings described in their manuscript to be freely available to other researchers, either 1. In a public repository, 2. Within the manuscript itself, or 3. Uploaded as supplementary information.

Additional Editor Comments (if provided):

Reviewers' comments:

Reviewer's Responses to Questions

**Comments to the Author**

1. If the authors have adequately addressed your comments raised in a previous round of review and you feel that this manuscript is now acceptable for publication, you may indicate that here to bypass the “Comments to the Author” section, enter your conflict of interest statement in the “Confidential to Editor” section, and submit your "Accept" recommendation.

Reviewer #1: (No Response)

Reviewer #2: All comments have been addressed

Reviewer #3: (No Response)

2. Does this manuscript meet PLOS Global Public Health’s publication criteria? Is the manuscript technically sound, and do the data support the conclusions? The manuscript must describe methodologically and ethically rigorous research with conclusions that are appropriately drawn based on the data presented.

Reviewer #1: Partly

Reviewer #2: Yes

Reviewer #3: Partly

3. Has the statistical analysis been performed appropriately and rigorously?

Reviewer #1: Yes

Reviewer #2: Yes

Reviewer #3: Yes

4. Have the authors made all data underlying the findings in their manuscript fully available (please refer to the Data Availability Statement at the start of the manuscript PDF file)?

Reviewer #1: No

Reviewer #2: Yes

Reviewer #3: No

5. Is the manuscript presented in an intelligible fashion and written in standard English?

Reviewer #1: Yes

Reviewer #2: Yes

Reviewer #3: Yes

6. Review Comments to the Author

Reviewer #1: Thank you for submitting your work on depressive symptomatology and hypertension. I read your work with great interest, as these are areas of growing concern globally, and as your introduction demonstrates, are issues of high concern in Kosovo. With revisions to the work, I’m sure there are many academics, and primary care and public health practitioners, who would find your study to be of great importance.

Overall, I feel the introduction, methods, and results are done to a high quality and are reflective of the care and consideration put into the study. Unfortunately, I feel there are several areas of the discussion section that need further revision, and additionally there are missing areas of the discussion section which I feel would be of importance to the reader. I appreciate a discussion section can be more difficult when the majority of findings are negative or non-significant, however, as you surely agree negative findings are of equal importance to the research community and I’m sure with revision of the discussion section there is much we can learn from your results.

My main concern with the discussion section is that it presents a single explanation (with regards to associations with people with depressive symptoms) for the paper’s findings regarding hypertension diagnosis (increased healthcare utilisation), and a single explanation for attenuated blood pressure (monoamine theory of depression). I feel both explanations are speculative, and although they are explained, they receive minimal evidence or justification from either the study’s findings, or other literature. There are a huge range of possible explanations and confounding factors for why depressive symptoms may be associated with slightly lower blood pressure in the first year of diagnosis, and why there may be an association with different rates of diagnosis and control of blood pressure. Broadly these could be biological, psychological, social and healthcare types of factors, and within each category there could be many explanations or confounding factors. There may be a variety of factors at play rather than a single explanation. It is also notable as well that the authors single out their inclusion of healthcare utilisation as a covariate as a possible reason for the non-significance of several findings, but do not grant this privilege to any other covariates (could smoking or physical inactivity equally exist along the causal pathway?). It will likely be unclear to the reader why the reasons given have been singled out, and a broader discussion would be of greater interest. In general, I would urge caution with some of the suggestions or conclusions proposed in the discussion, many of which are purely speculative and cannot be adequately justified by the results.

Additionally, the discussion section would benefit from suggesting implications for clinicians or policymakers. There is reference to the work being of epidemiological interest, and in the conclusion section, of the need for larger studies. Please include discussion of what implications there are for the work for primary care providers and for those in public health. Additionally, other than conducting larger studies, how else should further research in this area be directed based on the experience of this study?

Some of the methods could be enhanced by further discussion or explanation to the reader. I wasn’t personally familiar with the method used for censored normal regression in blood pressure follow-up over time, and it would be useful to explain why this was used and what effect it had on the results. Use of censored regressions does likely effect how the results should be interpreted. The models also include a relatively high number of covariates, and there is likely overlap between many of the covariates (e.g. alcohol consumption and nutrition). Could this have affected the ability of the regression models to find significant changes?

Finally, the section on weaknesses/limitations omits possible concerns about the study which should at the very least be stated. One example would be around how the study population may not represent a general population, and therefore may mean the results are not fully generalisable. For instance, there are several areas of possible selection bias in the study population: firstly, the vast majority of those eligible declined to participate; secondly, a large proportion were lost to follow-up; thirdly, within the paper it is not stated where or how patients were approached for recruitment (although reference is made to a study protocol elsewhere). It was not clear to me if the study itself also affected the rate at which people attended their primary care provider. For instance, how linked were the study visits to the primary care provided (either in selection or follow-up), and what happened if a participant had a high blood pressure reading detected in a study follow-up which was not previously known about? Additionally, this would be a good place to discuss the limitations of the study size, and the length of follow-up, which are alluded to elsewhere in the paper.

I additionally note that the data is not routinely made publicly available, although the appropriateness of this matter would be better addressed by the editor.

Line 53: Please be more precise than this in describing the evidence for depression and CVD as not all readers may be familiar with it, and it is of very high importance to your study. It would be useful to state briefly what the proposed relationship between depression and CVD is (e.g., if it’s a risk factor and in what direction), and if this has been demonstrated at the level of primary studies, or meta-analyses etc. This would aid readers in understanding the high importance of your research.

Lines 60-61: Whilst generally accepted that hypertension is an important risk factor for CVD, I feel you should include a reference if stating it is the most important of all risk factors.

Line 83: It seems significant that the study was conducted in primary care centres that do not offer mental health services. In your introduction (lines 67-69) you reference work which suggested that increased healthcare utilisation by those seeking help for depression might lead to higher hypertension diagnosis. Could this aspect of the primary care centres have affected your results in anyway? I could not see this referenced to in the discussion section.

Lines 96 & 98: It is not fully clear what is meant by ‘follow-up 3,’ is this meant to be the third follow-up visit? It’s not stated elsewhere how many follow-up visits occur or in what timeframe they occur. It may be worth rewording this, and further explaining the timeline of follow-up.

Lines 176 & 179: Both lines feature clauses in brackets, which are missing closed brackets.

Line 184: Please remove ‘a’ as you are referring to the plural form of multiple logistic regression models.

Line 202 & 230-233: Whilst you make good points here, commenting on the significance of your results, and comparison to other literature, would be better suited to the discussion section.

Line 242-248: There are many possible explanations for why those with moderate-to-severe depressive symptomatology may have an association with higher rates of hypertension diagnosis, and/or blood pressure attenuation. Your research has observed (mostly non-significant) tendencies but it has not investigated the reasons behind them. I think it is premature to single-out explanations, especially without reference to supporting literature or discussion of other plausible mechanisms. It is also not clear if your research findings definitively support any of the proposed explanations or why else they have been selected. It was also not clear from the paper how patients were recruited, and if those who were recruited had a higher tendency to healthcare-seeking than a baseline population of depressed patients (e.g. were patients recruited in a primary care centre they were already choosing to attend?). I don’t feel the study as it’s currently explained supports the suggestion that healthcare utilisation is primarily responsible for increased diagnosis. Furthermore, many of the findings were non-significant, which you only state by implication (‘only the findings with attenuated diastolic blood pressure met statistical significance’), instead of making this more explicit.

Lines 252-257: There is a huge variety of proposed theories and mechanisms for depression, and neurochemical explanations in particular are currently under quite intense scrutiny. If you wish to discuss this in relation to your hypertension findings, I feel it requires further explanation and justification to the reader. Or alternatively, making more explicit that there is a much wider field of possible explanations for the pathogenesis of depression (some of which may also have overlap with explanations for hypertension).

Lines 258-260: It is very interesting that there was minimal discrepancy between your change in systolic and diastolic blood pressure findings, between minimally- and fully-adjusted models. Perhaps this reflects relationships between your ‘minimal’ covariates (e.g. gender, educational attainment) and your ‘maximal’ covariates (e.g. smoking, physical activity).

Lines 271-275: It might be helpful to further justify this within the context of primary care in Kosovo. For instance, is it standard practice in Kosovo to routinely assess patients with depression for hypertension? This may not be standard practice globally in routine mental health primary care visits, and it would be useful to highlight its practical importance to clinicians.

Lines 266-268: ‘We found suggestive evidence that initially normotensive people with depressive symptoms were more likely to have hypertension detected by a general practitioner if it developed in comparison to non-depressed’. I think it would be best to omit ‘if it developed,’ as I don’t believe your study comparatively reported new hypertension which was undiagnosed in primary care (did your study follow-ups detect any patients with hypertension that had been missed by their primary care visits?). It would be more accurate just to state that normotensive people with depressive symptoms were more likely to have had hypertension diagnosed later on in follow-up. It would be interesting to know if there was a higher rate of undiagnosed hypertension in either group.

Lines 273-275: This is an excellent point, however you do include many covariates in your models, all of which could contain partial confounding factors or explanations for the relationship between depression symptoms and hypertension diagnosis/control. As before, it is not fully clear to me your justification for only choosing to discuss healthcare utilisation.

Lines 276-286: I would omit this paragraph. In proposing psychosomatic complaints as a basis for your findings, you are asking the reader to accept many steps, namely that i) those with depression attend their primary care provider more frequently; ii) a significant proportion or majority of these visits are for physical complaints and not depression; iii) the majority of these physical complaints are psychosomatic in nature; iv) all patients presenting with psychosomatic complaints have a blood pressure reading taken; v) any blood pressure reading beyond 140/90mmHg is treated as a separate issue and triggers a diagnostic pathway for hypertension. Furthermore, you support this only with a single reference to an observational study. Your final sentence, is a simplified statement on a very complex issue in primary care (and made with reference to an article from 1990; great strides have been made in primary care in the intervening decades in understanding psychosomatic illness and mental health). It’s quite a significant leap in my view to start proposing this a reason for your findings.

Lines 288-299: To refine this paragraphs readability, please revise the use of the word ‘however’ at the beginning of 3 different sentences.

Lines 292-294: You make a very interesting point here regarding the relationship between ethnic groups and hypertension control. I note that there is a significantly higher proportion of people of Roma, Ashkali, Egyptian, and other ethnicities represented in your group with moderate to severe depression symptoms. I’m sure this would be of great interest to the reader, particularly given that it is more unique to the setting of Kosovo. Could you expand further on this point, or is there other literature from Kosovo that may further expand this point?

Lines 310-311: I think you should make clear in the methods section that there was this degree of loss to follow-up as it does affect the reader’s interpretation of your study. Furthermore, you should state in the methods section how you handled subjects lost to follow-up.

Lines 312-314: It is interesting that you found many of those lost to follow-up, stopped attending due to Covid. Could this of affected the selection of your patient population, for instance were only healthier or younger participants more likely to accept the risk of coming for follow-up? Could this population of had a difference in the sequelae of a hypertension diagnosis?

Line 317-319: Firstly, I think the conclusion about age-related diastolic blood pressure change should be further clarified as something only detected within around a year of follow-up. Secondly, even though you state it is only ‘suggestive,’ I don’t think it would be appropriate to include the findings on detection and hypertension control as conclusions, given they were non-significant.

Lines 318-320: This is far too general a statement, I don’t think your study demonstrated definitively that there was no relationship between depression, hypertension, and CVD risk. I would omit reference to CVD risk in your conclusion, as your study doesn’t directly investigate incidence or risk of CVD.

Reviewer #2: (No Response)

Reviewer #3: ..

7. PLOS authors have the option to publish the peer review history of their article (what does this mean?). If published, this will include your full peer review and any attached files.

**Do you want your identity to be public for this peer review?** For information about this choice, including consent withdrawal, please see our Privacy Policy.

Reviewer #1: No

Reviewer #2: No

Reviewer #3: No

---

## [Editor Report · Decision Letter 2]

16 Jan 2023

PGPH-D-22-00718R2

Prospective association between depressive symptoms and blood pressure related outcomes in Kosovo

Dear Dr. Probst-Hensch,

Thank you for submitting your manuscript to PLOS Global Public Health. After careful consideration, we feel that it has merit but does not fully meet PLOS Global Public Health’s publication criteria as it currently stands. Therefore, we invite you to submit a revised version of the manuscript that addresses the points raised during the review process.

Specifically: we are grateful for your detailed responses to the comments from Reviewer #1, but it appears that your Response to Reviewers file does not address the comments raised by Reviewer #2 (attached for reference). Before we can proceed with further consideration of your manuscript, please revise your Response to Reviewers file to address the comments from Reviewer #2, as well as your manuscript file if necessary.

We look forward to receiving your revised manuscript.

Kind regards,

Hugh Cowley

Staff Editor

Journal Requirements:

If you did not receive any funding for this study, please simply state: “The authors received no specific funding for this work."
---

## [Editor Report · Decision Letter 3]

14 Mar 2023

Prospective association between depressive symptoms and blood pressure related outcomes in Kosovo

PGPH-D-22-00718R3

Dear Prof. Dr. Probst-Hensch,

We are pleased to inform you that your manuscript 'Prospective association between depressive symptoms and blood pressure related outcomes in Kosovo' has been provisionally accepted for publication in PLOS Global Public Health.

Best regards,

Paolo Angelo Cortesi, PhD

Academic Editor